# Susceptibility of *Aedes aegypti* Larvae to Temephos and Fenitrothion in Niamey (Niger) and Ouagadougou (Burkina Faso), Two West African Cities Recently Affected by Dengue

**DOI:** 10.3390/insects16090870

**Published:** 2025-08-22

**Authors:** Abdoul-Aziz Maiga, Aboubacar Sombié, Nicolas Zanré, Rahmatoulaye Maiga, Ibrahim Maman Laminou, Ali Doumma, Antoine Sanon, Athanase Badolo

**Affiliations:** 1Laboratoire d’Entomologie Fondamentale et Appliquée, Université Joseph KI-ZERBO, Ouagadougou 03 BP 7021, Burkina Faso; aboubacarsombie@yahoo.fr (A.S.); nicozanre12@gmail.com (N.Z.); rahmaiga2016@gmail.com (R.M.); sanonant@gmail.com (A.S.); 2Département des Sciences et Technologies, Université Moderne des Sciences de la Santé, Niamey, Niger; 3Unité de Paludologie et d’Entomologie Médicale, Centre de Recherche Médicale et Sanitaire, Niamey BP 10887, Niger; lamine.cermes@gmail.com; 4Département de Biologie, Faculté des Sciences et Techniques, Université Abdou Moumouni, Niamey BP 10896, Niger; doumma@yahoo.com

**Keywords:** *Aedes aegypti*, insecticide resistance, larvicides, organophosphates, vector control, dengue, West Africa

## Abstract

This study assessed the susceptibility profile of the immature stages of *Ae. aegypti*, the primary arbovirus vector to temephos and fenitrothion in urban settings of Niamey and Ouagadougou. The work concentrations of the larvicides were prepared by diluting technical grades of temephos and fenitrothion in acetone. Larval bioassays were performed according to World Health Organization protocol. The vector populations from both cities were found susceptible to both temephos and fenitrothion, suggesting that both insecticides could be used for larval source management as dengue control response in the two localities. The reduced susceptibility patterns to temephos exhibited by *Ae. aegypti* larvae of Ouagadougou suggest that a follow up monitoring of the larval susceptibility is required.

## 1. Introduction

*Aedes aegypti* is responsible for the spread of several viral diseases, including dengue, yellow fever, chikungunya, and Zika [1]. Diseases transmitted by *Ae. aegypti* and *Ae. albopictus* are increasingly becoming a major challenge in Africa, with an estimated population at risk of 1.24 (1.24–1.24) billion for dengue, chikungunya, and Zika and of 1.14 (1.14–1.15) billion for yellow fever [2]. The resurgence of these diseases can be attributed to unplanned urbanization, climate change [3], and increased international exchanges [4]. This complicates control efforts, especially in resource-limited countries such as those in West Africa. In 2023, the highest number of arbovirus outbreaks were recorded in Senegal (Dengue, Chikungunya, Zika, Yellow Fever, Crimean–Congo hemorrhagic fever, and West Nile), Mali (dengue, chikungunya, and Zika) and Burkina Faso (dengue and chikungunya) [5]. Human serological investigations conducted among febrile patients in Niamey in 2023 revealed seven autochthonous cases of the dengue virus (DENV-1 and DENV-3) but no Chikungunya (CHIKV) and Zika (ZIKV) [6]. A previous study conducted in 1965 revealed sero-circulation of CHIKV, ZIKV, and West Nile Virus (WNV) among humans in Tera, located 185 km from Niamey [7]. In Ouagadougou, an unprecedented epidemic occurred in 2023, resulting in 70,433 cases and 709 deaths [8], alongside a chikungunya outbreak with 89 confirmed cases, though no fatal cases were reported [9].

Due to the absence or limited access to effective vaccines and lack of specific medicines targeting DENV, CHIKV, and ZIKV [10], vector control remains the cornerstone of disease prevention and control [11]. Larviciding is an effective preventative method of arboviruses, helping to reduce vector density and infection risk by targeting larvae before they develop into adult mosquitoes [11]. It also aids in managing insecticide resistance [12] by minimizing the resistance selection among adult mosquitoes. A number of studies have shown that approaches targeting immature stages of *Ae. aegypti* are more effective over time than those targeting adults alone, though a combination of both yields more significant impact on entomological and epidemiological indicators [13].

Chemical larvicides that are commonly used in *Aedes* mosquito control are organophosphates and insect growth regulators [13]. Temephos is widely used globally [14], whereas fenitrothion has been used extensively in Thailand [15] and only to a limited extent in Indonesia [16]. Organophosphates act by inhibiting detoxifying enzymes known as cholinesterase, leading to overstimulation of nerve endings due to the accumulation of acetylcholine [17]. In contrast, growth regulators disrupt hormone expression in larvae and pupae, interrupting metamorphosis process and either preventing adult emergence or resulting in deformed adults [17].

A study conducted in Ouagadougou in 2016 revealed *Aedes aegypti* larval susceptibility to temephos and fenitrothion [18]. Given the expanding geographical range and increasing intensity of dengue infections in the country [8], it is essential to provide updated data on the susceptibility of immature stages to these larvicides. In Niger, no data are currently available on the efficacy of larvicides against immature stages of *Ae. Aegypti*. However, data showing resistance to pyrethroid insecticides in adult mosquitoes have been available since last year [19]. This study aims to fill this gap by providing a baseline data on the susceptibility of *Ae. aegypti* larvae to organophosphates (temephos and fenitrothion) in Niamey and updated data from Ouagadougou in order to assess how mosquito responses to these insecticides evolve with time across different localities.

## 2. Materials and Methods

### 2.1. Sampling Sites, Period, and Methods

*Ae. aegypti* eggs and larvae were sampled, respectively, at Yantala and Lamordé, two neighborhoods in Niamey (13°31′17″ N, 2°6′19.001″ E), the capital city of Niger, and from Zongo in Ouagadougou (12°21′56.38″ N, 1°32′1.97″ W), the capital city of Burkina Faso. Niamey has a surface area of 255 km^2^ and a Sahelian climate, with annual rainfall ranging from 300 to 600 mm. Ouagadougou, covering an area of 520 km^2^, is situated in the Sudano-Sahelian zone, with an average annual rainfall of 600 to 900 mm.

Yantala is an urban neighborhood of Niamey city on the left bank (Rive Gauche) of the Niger river, whereas Lamordé is a peri-urban area located on the right bank (Rive Droite). Zongo is a peri-urban locality in the western part of Ouagadougou.

In Niamey eggs were collected using ovitraps during the 2022 rainy season (June–October). The ovitraps were placed outdoors and around human dwellings for 72 h. Then, the egg collection papers were retrieved and dried at room temperature for no more than 24 h, packaged in plastic bags, and transported to the insectarium of the Laboratoire d’Entomologie Fondamentale et Appliquée, Joseph Ki-Zerbo University, Ouagadougou, Burkina Faso. In Ouagadougou, larvae were collected in 2023 during the rainy season (May–October) in three types of containers: drums, metallic vessels, and plastic containers.

All households included in the study gave their oral consent prior to any sampling activities. The importance of the entomological collections and the usefulness of the study for public health decision-making were clearly explained.

### 2.2. Eggs Hatching and Larvae Rearing

Eggs from Niamey were hatched by submerging the papers in trays containing distilled water under insectarium conditions at 27.7 ± 1.4 °C temperature and 79.1 ± 5.5% relative humidity, with a photoperiod of 12:12 (Light:Dark). After hatching, the larvae were fed with Tetramin^®^ (Tetra Fish products, Melle, Germany). The water in the trays was changed every two days. Larvae collected from the field in Ouagadougou were reared in the same insectary. The larvae were monitored until they reached the advanced stages of L3 and early L4 stages, at which point the bioassays were conducted. First generation (F1) from field-collected mosquitoes in Ouagadougou and Niamey were used for bioassay tests. The Liverpool strain was used as a susceptible reference strain.

### 2.3. Larvicides and Concentrations Used

Technical grade of temephos (Abate, AccuStandard, New Haven, CT, USA) and fenitrothion (FUJIFILM Wako Pure Chemical Corporation, Osaka, Japan) were used at the following concentrations: 0.441, 0.738, 1.038, 1.479, 2.217, and 3 mg/L for temephos and 0.441, 0.591, 0.738, 1.107, 1.479, and 2.217 mg/L, for fenitrothion. These concentrations were prepared by diluting the technical grade of the two insecticides in acetone.

### 2.4. Larvicides Susceptibility Tests

The susceptibility bioassays were performed following the World Health Organization (WHO) protocol [20]. One milliliter (1 mL) of each insecticide solution was added to 99 mL of distilled water. The resulting concentrations were as follows: 0.0044, 0.0073, 0.0103, 0.0147, 0.0221, and 0.03 mg/L for temephos and 0.0044, 0.0059, 0.0074, 0.0111, 0.0148 and 0.0222 mg/L for fenitrothion, in a total volume of 100 mL. Late third to early fourth instar of *Aedes aegypti* larvae were used, in batches of 20–25 individuals. For each concentration, four replicates and one control were carried out. In the control tray, 1 mL of acetone (the solvent) was added to 99 mL of distilled water to make up the total volume to 100 mL. After a 24 h exposure period, larval mortality was recorded. Each insecticide test was repeated three times on different days following the WHO protocol [20].

### 2.5. Data Analysis

For each concentration, the mortality rate was calculated by dividing the number of dead or moribund larvae by the total number of larvae tested and then multiplied by 100. In cases where pupation occurred, the number of pupae was subtracted from the total number of larvae to obtain the exact number larvae used. If the mortality in control ranged between 5% and 20%, the Abbott correction was applied. Tests were discarded if the number of pupae exceeded 10% [20].

R software version 4.5.0 [21] was used for the determination of lethal dose (LD_50_), which corresponds to the concentration killing 50% of exposed larvae for each insecticide. The resistance ratio (RR) was calculated by dividing the LD_50_ of the field population to those of the susceptible Liverpool strain according to the following formula:

The resistance Ratio (RR) = LD_50_ field *Aedes* population/LD_50_ susceptible strain (Liverpool). The interpretation of bioassay results followed the WHO protocol [20]. A mosquito population was considered susceptible when RR < 5; the population was moderately resistant for RR between 5 and 10 and highly resistant when RR > 10.

## 3. Results

Both larval populations of *Ae. aegypti* from Niamey, Lamordé (Rive Droite) and Yantala (Rive Gauche), were found to be susceptible to temephos and fenitrothion (Table 1 and Appendix A). For temephos, the LD_50_ values were 0.0089 mg/L [0.0002–0.077] for Lamordé and 0.0113 mg/L [0.0004–0.079] for Yantala. The RR_50_ values were 2.36 and 3.00, while RR_95_ values were 5.23 and 4.75 for Lamordé and Yantala populations, respectively (Table 1). For fenitrothion, the LD_50_ values were 0.0075 mg/L [0.0004–0.04880] for Lamordé and 0.0109 mg/L [0.0008–0.0563] for Yantala (Table 1). The RR_50_ values were 0.62 and 0.91 for Lamordé and Yantala, respectively.

All three *Aedes aegypti* larval populations from different breeding containers in Ouagadougou—drums, metallic containers, and plastic containers—were also susceptible to both temephos and fenitrothion (Table 1 and Appendix A). For temephos, the LD_50_ values were 0.0059 mg/L [0.0002–0.0444] for drums, 0.0083 mg/L [0.0000–0.1492] for metallic containers, and 0.0068 mg/L [0.0002–0.0533] for plastic containers. The corresponding RR_50_ values were 1.56, 2.21, and 1.81, respectively. For fenitrothion, the LD_50_ values were 0.057 mg/L [0.0009–0.0217] for drums, 0.058 mg/L [0.0001–0.0515] for metallic containers, and 0.057 mg/L [0.0005–0.0311] for plastic containers. The corresponding RR_50_ values were 0.47, 0.48, and 0.47, respectively, across these mosquito larval habitats.

Although *Ae. aegypti* larvae populations from the two cities were found susceptible to temephos and fenitrothion, a trend of reduced sensitivity was observed in their response to larvicide exposure. When comparing populations from Lamordé (RD) in Niamey and Zongo in Ouagadougou, two sites with approximatively the same urbanization patterns, we surprisingly found that *Ae. aegypti* populations from RD were, respectively, 1.5, 1.03, and 1.3 times less susceptible to temephos than those from Zongo drums, metallic containers, and plastic containers. Similarly, they were 1.3 times less susceptible to fenitrothion when comparing their RR_50_. A similar pattern was found between RG and Zongo populations. *Ae. aegypti* populations from Yantala (RG) in Niamey were, respectively, 1.9, 1.3, and 1.6 times less susceptible to temephos compared to those from Zongo drums, metallic containers, and plastic containers in Ouagadougou. For fenitrothion, the same tendency was observed, with *Ae. aegypti* larval populations from Niamey at Yantala (RG) showing an RR_50_ that was 1.9 times higher than those from Ouagadougou.

## 4. Discussion

This study assessed the susceptibility profile of larval *Ae. aegypti* mosquitoes from two West African cities, Niamey (Niger) and Ouagadougou (Burkina Faso), to two organophosphate larvicides, temephos and fenitrothion. Overall, all the field mosquito populations tested were susceptible to the two larvicides with RR_50_ < 5. This provides evidence that these larvicides could be effectively used for preventing arboviruses transmitted by *Ae. aegypti* in these cities of West Africa.

In Niamey specifically, this study reports for the first time data on the susceptibility status of *Ae. Aegypti* larvae to temephos and fenitrothion, thus providing essential baseline data for monitoring evolution of *Ae. aegypti* susceptibility to these larvicides in the coming years. As Niger has never experienced any dengue outbreak or epidemic, *Ae. aegypti* has not been specifically targeted by vector control. However, it is likely that the intensity and the geographic range of DENV are underreported in the country due to the absence of active epidemiological monitoring and the frequent misdiagnosis of dengue as malaria, as they share similar symptoms [22]. This study’s results, alongside previous findings on *Ae. aegypti* adult mosquitoes from Niamey, which highlighted resistance to pyrethroids and susceptibility to carbamates and organophosphates [19], provide evidence-based data to support chemical vector control strategies in the city of Niamey by using chemical insecticides. For larvicide deployment to be effective against *Ae. aegypti* mosquitoes, further research is needed to characterize the types of breeding containers used by the vector. Furthermore, community sensitization and education on good practices to prevent *Ae.aegypti* proliferation in their domestic and peri-domestic environments could play a complementary role in preventing dengue and other arboviruses in the city of Niamey.

In Burkina Faso, this study confirms that *Ae. aegypti* populations from the semi-urban setting of Ouagadougou are still susceptible to temephos and fenitrothion as previously reported 9 years ago in a similar peri-urban locality. However, reduced susceptibility to temephos and quite similar susceptibility to fenitrothion were observed [18]. This trend is concerning, as it could signal the potential emergence of *Ae. aegypti* resistance to temephos in Ouagadougou. It is important to note that the vector has been targeted by vector control (outdoor space spraying) with organophosphate insecticides such as malathion and pirimiphos-methyl during the 2017 outbreak [18]. Further research is needed to assess how household use of insecticides could impact the emergence of resistance among urban *Ae. aegypti* populations. In the context of reemergence of DENV and the emergence of CHIKV in the county [8,9], insecticide susceptibility results combined with knowledge of *Ae. aegypti* breeding sites characteristics and typology in Ouagadougou [23] provide critical information for outbreak preparedness and response.

The susceptibility of *Ae. aegypti* to temephos has also been reported in other West and Central African countries. In West Africa, Mauritania and Nigeria reported *Ae. aegypti* populations as more susceptible to temephos compared to those observed in this study, based on the RR_50_ comparisons [24,25]. In Central Africa, countries such as Cameroon [26,27], Gabon [26], Central African Republic [28], and the Republic of Congo [29] also reported higher susceptibility levels according to RR_50_ values. However, confirmed cases of *Aedes aegypti* resistance to temephos have been recorded in Cape Verde [30]. Globally, *Ae. aegypti* resistance to temephos has been largely reported in dengue-endemic countries of Latin America, including Brazil [31], Colombia [32], and Costa Rica [33], as well as in Asia, particularly in Thailand [34] and Indonesia [16]. With regard to fenitrothion, few studies have assessed *Ae. aegypti* larval resistance in Africa. To date, data are available only for Ouagadougou, where vector populations have been found to be susceptible [18]. To our knowledge, *Ae. aegypti* larval resistance to fenitrothion has not yet been recorded in Africa. Elsewhere, susceptibility to fenitrothion has been observed in Indonesia [16] and Costa Rica [33], although a case of resistance was also reported in Indonesia [16].

## 5. Conclusions

This study reported the susceptibility status of *Ae. aegypti* larvae populations to temephos and fenitrothion in two cities of West Africa, Niamey (Niger) and Ouagadougou (Burkina Faso). *Ae. aegypti* larval populations from both cities were found to be susceptible to these larvicides, supporting their potential use in arbovirus outbreaks response efforts in these major urban centers. However, the reduced susceptibility observed among *Ae. aegypti* larvae in Ouagadougou highlights the need to assess insecticide efficacy as part of routine surveillance. This is the first study reporting *Ae. aegypti* larval mosquitoes’ susceptibility to temephos and fenitrothion in Niger.

## Figures and Tables

**Table 1 insects-16-00870-t001:** Summary of dose–response values from larval bioassays (LC_50_ and LC_95_), mg/L, with 95% confidence limits (95% CI).

Insecticide	*Aedes aegypti* Population		LC (95% CI) (mg/L)		RR		Resistance Status
			LC_50_	LC_95_	RR_50_	RR_95_	
Temephos		Reference strain	0.0038 (0.0000–0.1524)	0.0075 (0.0000–0.2416)	1.00	1.00	S
	Niger	Lamordé (Rive Droite)	0.0089 (0.0002–0.0767)	0.0395 (0.0013–0.2402)	2.36	5.23	S
		Yantala (Rive Gauche)	0.0113 (0.0004–0.0788)	0.0358 (0.0018–0.1945)	3	4.75	S
	Burkina Faso	Zongo drums	0.0059 (0.0002–0.0444)	0.0294 (0.0016–0.1591)	1.56	3.89	S
		Zongo metallic	0.0083 (0.0000–0.1492)	0.0223 (0.0000–0.2954)	2.21	2.95	S
		Zongo Plastic	0.0068 (0.0002–0.0533)	0.0357 (0.0018–0.1956)	1.81	4.75	S
Fenitrothion		Reference strain	0.0121 (0.0001–0.1158)	0.0257 (0.0004–0.2031)	1.00	1.00	S
	Niger	Lamordé (Rive Droite)	0.0075 (0.0004–0.04880	0.0159 (0.0010–0.0893)	0.624	0.62	S
		Yantala (Rive Gauche)	0.0109 (0.0008–0.0563)	0.0234 (0.0022–0.1054)	0.91	0.91	S
	Burkina Faso	Zongo drums	0.0057 (0.0009–0.0217)	0.0154 (0.0030–0.0516)	0.47	0.60	S
		Zongo metallic	0.0058 (0.0001–0.0515)	0.0188 (0.0006–0.1289)	0.48	0.73	S
		Zongo Plastic	0.0057 (0.0005–0.0311)	0.0130 (0.0013–0.0616)	0.47	0.51	S

## Data Availability

The original contributions presented in this study are included in the article/Appendix A. Further inquiries can be directed to the corresponding authors.

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
