# Peer review of "Susceptibility of Aedes aegypti Larvae to Temephos and Fenitrothion in Niamey (Niger) and Ouagadougou (Burkina Faso), Two West African Cities Recently Affected by Dengue"

_insects, 2025, doi:10.3390/insects16090870_

Round 1

Reviewer 1 Report

Comments and Suggestions for Authors

Thank you for your manuscript entitled Susceptibility of Aedes aegypti larvae to temephos and fenitrothion, in Niamey (Niger) and Ouagadougou (Burkina Faso), two West African cities recently affected by dengue. It is interesting but a few corrections should be done before its publication

You should include one to two sentences to describe the methods in the simple summary.

Lines 45–46: You should verify the estimated population at risk for each arbovirus.

Line 49: You should include the number of Aedes-borne outbreaks for each country.

Lines 73–76: You should clarify the meaning of your sentences. It looks like there is an extra “by.”

In your Methods section, include a description of adult and/or larval identification. In addition, you should consistently use the past tense in the Methods section. Many sentences, such as those in lines 112–114, are currently written in the present tense.

Line 111: The company and country that supplied Tetramin should be included.

Lines 114–116: You should clarify the mosquito generation (F₀, F₁, or F₂) used in your bioassay tests.

You should include a table or graph showing the mortality rate obtained from each treatment or experiment, either in your supplementary data or in the results section. In addition, the text in your results should summarize the main findings and/or provide a biological explanation, rather than repeating the exact content of Table 1.

Lines 150–151: Table 2 is missing, or did you mean Table 1?

Lines 155–156: You should clarify the meaning of your sentence.

In your discussion, you should include a few sentences about the resistance mechanisms observed in your mosquito populations.

Comments on the Quality of English Language

There are many sentences with extra words, such as "by" and "of." In addition, some sentences in your Methods section are written in the present tense. You should ensure consistency by using the past tense throughout the Methods section.

Author Response

Reviewer: You should include one to two sentences to describe the methods in the simple summary.

Authors: OK done.

Reviewer: Lines 45–46: You should verify the estimated population at risk for each arbovirus.

Authors : The estimated population at risk was given together for dengue, chikungunya and Zika (probably because they share the same characteristics of area of risk), and for yellow fever alone apart, please see table 1 of the paper: https://doi.org/10.1038/s41467-025-58609-5  

Reviewer: Line 49: You should include the number of Aedes-borne outbreaks for each country.

Authors : Ok done

Reviewer: Lines 73–76: You should clarify the meaning of your sentences. It looks like there is an extra “by.”

Authors : Ok reformulated.

Reviewer: In your Methods section, include a description of adult and/or larval identification. In addition, you should consistently use the past tense in the Methods section. Many sentences, such as those in lines 112–114, are currently written in the present tense.

Authors : Ok corrected.

Reviewer: Line 111: The company and country that supplied Tetramin should be included.

Authors : OK provided.

Reviewer: Lines 114–116: You should clarify the mosquito generation (F₀, F₁, or F₂) used in your bioassay tests.

Authors : Ok done

You should include a table or graph showing the mortality rate obtained from each treatment or experiment, either in your supplementary data or in the results section. In addition, the text in your results should summarize the main findings and/or provide a biological explanation, rather than repeating the exact content of Table 1.

Reviewer: Lines 150–151: Table 2 is missing, or did you mean Table 1?

Authors : I meant table 1

Reviewer: Lines 155–156: You should clarify the meaning of your sentence.

Authors : Ok reformulated

Reviewer: In your discussion, you should include a few sentences about the resistance mechanisms observed in your mosquito populations.

Authors : Ok done

Reviewer : Comments on the Quality of English Language

There are many sentences with extra words, such as "by" and "of." In addition, some sentences in your Methods section are written in the present tense. You should ensure consistency by using the past tense throughout the Methods section.

Authors : Ok corrected.

Reviewer 2 Report

Comments and Suggestions for Authors

While the study presents data on Aedes aegypti larval susceptibility to temephos and fenitrothion from two West African cities, the overall scientific novelty and clarity of presentation are limited. Several concerns reduce its contribution to the existing literature.

Major Concerns

  1. Lack of Novelty

The study confirms that Aedes aegypti populations remain susceptible to temephos and fenitrothion—an observation that has already been well documented in many previous studies from Africa and other regions.Even within the manuscript, the authors cite past studies from Burkina Faso (2016) and Ouagadougou, as well as multiple other African countries, all reporting similar susceptibility outcomes.Therefore, no new or unexpected findings emerge from this study. The conclusions largely reiterate known trends without advancing mechanistic insights, resistance evolution patterns, or resistance management strategies. The author has to justify the novel information this study adds to already available literature.

  1. Data Presentation

Results are presented only in tabular format, which is not reader-friendly and lacks visual clarity. Dose-response relationships and resistance ratios would have benefited from graphical representation (e.g., probit mortality curves, bar charts, or RR plots) to visually summarize trends across sites and larvicides. Tables are dense, difficult to follow, and do not allow for easy comparisons between populations or larvicides.

  1. Inconsistent and Confusing Notation

The manuscript shows inconsistent use of decimal points and commas (e.g., “0,0089 mg/L” and “0.0089 mg/L” are both used), which is confusing and unprofessional, especially for an international audience. A consistent formatting convention (preferably SI-standard decimal points) should be applied throughout the manuscript.

  1. Limited Geographic or Ecological Scope

Although the authors mention urbanization patterns and different larval habitats, the ecological factors influencing susceptibility (e.g., breeding site type, insecticide exposure history, household practices) are not explored or statistically analyzed. The comparisons made between sites (e.g., Lamordé vs. Zongo) are purely descriptive, lacking any robust statistical testing or discussion of environmental variables.

Author Response

Major Concerns

  1. Lack of Novelty

 Reviewer: The study confirms that Aedes aegypti populations remain susceptible to temephos and fenitrothion—an observation that has already been well documented in many previous studies from Africa and other regions.Even within the manuscript, the authors cite past studies from Burkina Faso (2016) and Ouagadougou, as well as multiple other African countries, all reporting similar susceptibility outcomes.Therefore, no new or unexpected findings emerge from this study. The conclusions largely reiterate known trends without advancing mechanistic insights, resistance evolution patterns, or resistance management strategies. The author has to justify the novel information this study adds to already available literature.

Authors : Thank you for your valuable comments. The novelty of this study lies in the reduced susceptibility of Ae. aegypti larvae from Ouagadougou to temephos, compared to baseline data from 2016 (Badolo et al., 2019), and we believe this result warrants reporting. We were surprised to observe similar results for temephos susceptibility in Niamey. Given that both countries have experienced dengue cases, and Burkina Faso is subject to outbreaks, such a reduction in susceptibility to temephos should be reported.

  1. Data Presentation

Reviewer: Results are presented only in tabular format, which is not reader-friendly and lacks visual clarity. Dose-response relationships and resistance ratios would have benefited from graphical representation (e.g., probit mortality curves, bar charts, or RR plots) to visually summarize trends across sites and larvicides. Tables are dense, difficult to follow, and do not allow for easy comparisons between populations or larvicides.

Authors : The way within we presented our data is commonly used (Kamgang et al, 2011; Ngoagouni et al, 2016; Yougang et al, 2020 and Haidy-Massa et al, 2024).

  1. Inconsistent and Confusing Notation

Reviewer: The manuscript shows inconsistent use of decimal points and commas (e.g., “0,0089 mg/L” and “0.0089 mg/L” are both used), which is confusing and unprofessional, especially for an international audience. A consistent formatting convention (preferably SI-standard decimal points) should be applied throughout the manuscript.

Authors : Thank you for your comments. Ok corrected.

  1. Limited Geographic or Ecological Scope

Reviewer: Although the authors mention urbanization patterns and different larval habitats, the ecological factors influencing susceptibility (e.g., breeding site type, insecticide exposure history, household practices) are not explored or statistically analyzed. The comparisons made between sites (e.g., Lamordé vs. Zongo) are purely descriptive, lacking any robust statistical testing or discussion of environmental variables.

Authors : Thank you for your valuables comments. We have improved accordingly. However, we have evocated for each city if there has been or not vector control intervention that specifically targeted Ae. aegypti mosquitoes. There is no significant difference between mosquitoes responses from a site to another, just by looking at to the confidence interval values of different RR50.

Round 2

Reviewer 1 Report

Comments and Suggestions for Authors

The authors did a good job addressing most of the comments, but they forgot to include a table summarizing the mortality rate for each experiment, as I previously requested. Without this, it is difficult to properly evaluate the results of this manuscript.

Author Response

Comment: the authors did a good job adressing most of the comments, but they forgot to include a table summarizing the mortality rate of each experiment, as I previously requested. Without this, it is difficult to properly evaluate the results of this manuscript.

Response: Ok, provided as supplementary data (S1 for Niamey and S2 for Ouagadougou)

Reviewer 2 Report

Comments and Suggestions for Authors

The authors have adequately addressed all the concerns raised during the review process. They have provided satisfactory clarifications and made the necessary revisions to improve the manuscript. Therefore, I recommend the manuscript for publication.

Author Response

Comments: The authors have adequately adreesed all the concers raised during the reviex process. They have provided satisfactory clarifications and made the necessary revisions to improve the manuscript. Therefore, I recommed the manuscript for publication.

Response: Thank you very much for your valuable comments that helped to improve the quality of the manuscript.

Round 3

Reviewer 1 Report

Comments and Suggestions for Authors

The supplementary data should be indicated in the manuscript. The manuscript mentions no supplementary data, which is not accurate.

Author Response

Comments: The supplementary data should be indicated in the manuscript. The manuscript mentions no supplementary data, which is not accurate.

Response: We have indicated supplementary data in the manuscript as S1 and S2. About to introduce these tables in the manuscript, seriously we think that they should not be included in the text, because they are raw data, and the table 1 is from these raw data. We think it is better to notify that supplementary data exists for the manuscript at the bottom of the manuscript.